# Public Engagement and Neurology: An Update

**DOI:** 10.3390/brainsci11040429

**Published:** 2021-03-28

**Authors:** Luigi Lavorgna, Francesco Brigo, Sabrina Esposito, Gianmarco Abbadessa, Maddalena Sparaco, Roberta Lanzillo, Marcello Moccia, Matilde Inglese, Luca Bonfanti, Francesca Trojsi, Emanuele Spina, Antonio Russo, Pasquale De Micco, Marinella Clerico, Gioacchino Tedeschi, Simona Bonavita

**Affiliations:** 1Division of Neurology, Department of Advanced Medical and Surgical Sciences, AOU—University of Campania “Luigi Vanvitelli”, P.zza Miraglia 2, 80138 Naples, Italy; luigi.lavorgna@policliniconapoli.it (L.L.); sabrina.esposito1@unicampania.it (S.E.); gianmarcoabbadessa@live.com (G.A.); ms.86@hotmail.it (M.S.); dottor.russo@gmail.com (A.R.); gioacchino.tedeschi@unicampania.it (G.T.); simona.bonavita@unicampania.it (S.B.); 2Department of Neurology, Hospital of Merano (SABES-ASDAA), 39012 Merano, Italy; dr.francescobrigo@gmail.com; 3Multiple Sclerosis Clinical Care and Research Centre, Department of Neuroscience, Reproductive Science and Odontostomatology, Federico II University, 80138 Naples, Italy; roberta.lanzillo@unina.it (R.L.); moccia.marcello@gmail.com (M.M.); emanuelespina499@hotmail.com (E.S.); 4Italy—OSPEDALE San Martino, IRCCS, University of Genova, 16132 Genoa, Italy; m.inglese@unige.it; 5Neuroscience Institute Cavalieri Ottolenghi (NICO), 10043 Orbassano, Italy; luca.bonfanti@unito.it; 6Department of Veterinary Sciences, University of Turin, 10095 Torino, Italy; 7Association of Engineers of Naples, 80121 Naples, Italy; ing.demicco@gmail.com; 8Clinical and Biological Sciences Department, University of Torino, 10124 Turin, Italy; marinella.clerico@unito.it

**Keywords:** public engagement, campaign, educational, awareness, fundraising, foundation, association, prevention

## Abstract

Background: Public engagement (PE) is defined as the involvement of “specialists who listen, develop their understanding, and interact with non-specialists in non-profit activities of educational, cultural, and social nature to engage the public in science-related matters”. The public health relevance of PE consists in building up a scientifically literate society, able to participate in and support scientific and technological developments and their implications for educational settings. Neurological disorders account for 35% of all diseases. PE could have a positive impact on the lives of people affected by neurological diseases. Method: This review evaluates the role of PE in dementia, stroke, epilepsy, multiple sclerosis, Parkinson’s disease, migraine, neurogenetics, and amyotrophic lateral sclerosis. Results and Conclusions: PE can provide accessible information, support research activities and prevention through appropriate lifestyles, and increase knowledge and awareness of neurological disorders, improving their diagnosis and treatment.

## 1. Introduction

As defined by the Higher Education Funding Council for England, HEFCE, in 2006, public engagement (PE) is the involvement of “specialists who listen, develop their understanding, and interact with non-specialists in non-profit activities of educational, cultural, and social nature to engage the public in science-related matters”.

Visible scientists, science communicators, journalists, writers, museum curators, interpreters, the press, and public relations professionals play a central, relevant role in PE, and can contribute to the dissemination of processes and results of scientific research (Table 1) [1].

The public health relevance of PE has recently been emphasized by the European Commission, which has highlighted its role in building up a scientifically literate society, able to participate in and support scientific and technological developments and their implications for educational settings [2].

According to the World Health Organization (WHO), neurological disorders account for 35% of all diseases in Europe [3]. A research model that positively affects people with neurological diseases and their caregivers requires support and implementation by all relevant and key stakeholders [4]. In this regard, the EU-funded MULTI-ACT project [5] has recently been developed to increase the impact of health research on people with neurological diseases by creating and implementing a new governance model based on the effective cooperation of all relevant stakeholders [6].

In this systematic review, we evaluate the role of PE in the main neurological disorders to identify issues to be improved and educational gaps to be filled.

## 2. Materials and Methods

In this review, we have included articles specifically addressing the role of PE in the following neurological disorders or conditions: dementia, stroke, epilepsy, multiple sclerosis, Parkinson’s disease, migraine, neurogenetics, and amyotrophic lateral sclerosis. We have excluded articles on PE not related to the aforementioned neurological diseases. For the assessment of articles for possible inclusion, we adopted the definition of PE provided by the university funding agency of England as reported above [1]. To assist reviewers in the article selection, we provided some examples of PE (Table 1).

An advanced literature search in MEDLINE (accessed through PubMed) was conducted at the beginning of 2020 by two independent authors (G.A., S.E.). The search terms were “*(public engagement OR third mission) AND (motor neuron disease OR amyotrophic lateral sclerosis OR Alzheimer’s OR dementia OR multiple sclerosis OR epilepsy OR stroke OR Parkinson’s disease OR headache OR migraine OR neurology and genetics)*”. We ordered the results according to the “best match” filter. The search was restricted to articles from peer-reviewed journals published in English. No date restrictions were used. The repeatability of the data selection method between the 2 reviewers was tested using a stepwise searching approach. Starting from the research with the greatest number of results (“*(dementia OR Alzheimer’s) AND (public engagement OR third mission)*”), the first 100 results of 540 were independently evaluated by the two reviewers. Therefore, any disagreement was discussed to clarify selection criteria. Agreement on screening decisions was >95%, so the further selections regarding other combinations were performed by one reviewer only (G.A.). The reference lists of the included studies were manually searched to identify further relevant studies. A detailed flow chart of article selection is presented in Figure 1.

Figure legend:

Moreover, based on the PE activities formally identified for qualitative analyses by the Department of Advanced Medical and Surgical Sciences of the University of Campania “Luigi Vanvitelli”, we also carried out a search on Google for webpages, online communities, forums, sites, and institutional websites, using the following terms (even in combination): “Alzheimer’s disease”, “dementia”, “stroke”, “epilepsy”, “multiple sclerosis”, “Parkinson’s disease”, “migraine”, “headache”, “Huntington’s disease” “neurofibromatosis”, “muscular dystrophy”, “tuberous sclerosis”, “rare neurological disease”, “amyotrophic lateral sclerosis”, “public engagement” “awareness day”, “day”, “open day”, “fundraising”, “patients’ associations”, “research engagement”, “parent project”, “exhibition”, “No Profit Organization”, “engagement day”, “campaign”, “foundation”, “alliance”, “celebrities”, “viral event”, “non-profit organization”, “prevention”, “museum”, “sport”. Relevant results from the Google search were reported.

## 3. Results

### 3.1. Dementia

Worldwide, the number of people living with dementia (PLWD) is continuously increasing, and will reach an estimated 75.63 million in 2030 and 135.46 million in 2050 [7].

Routine care activities, coordination of healthcare services [8], and management of finances are often provided by family members of PLWD [9]. Hence, the engagement of the “patient–caregiver dyad” is essential to identify care and research priorities [10], and to develop more effective healthcare interventions [11].

To date, there is little data on PE in dementia, particularly on its effect on research recruitment, and the impact on laypeople and PLWD [12]. However, over the last few years, many popular initiatives and studies have been conducted to increase dementia awareness.

They can be grouped into four main categories of initiatives:

(1) Surveys highlighted some relevant issues regarding the well-being of caregivers and PLWD. For instance, the safety of people with dementia and management of the caregiving burden were identified as the main aims of caregivers [13]; conversely, patients with early dementia identified as primary goals the preservation of work engagement and family functions, the need to not be a burden on the family, and the need for community education [14].

(2) The “dementia-friendly” community was defined by Alzheimer’s Disease International (ADI) as “a place of culture in which people with dementia and their caregivers are empowered, supported and included in society, understand their rights and recognize their full potential” [15].

In 2012, WHO and ADI introduced this inclusive model of dementia in society, in which all patients can remain engaged in their usual everyday life for as long as possible. In 2016, ADI released a report identifying “dementia-friendly” communities in over 30 countries worldwide. A wide range of innovative programs were promoted, such as the “UK Dementia Friends” program and the “Netherlands’ Alzheimer Café”, with international recognition and divulgation [16].

(3) Public initiatives and caregiver training programs: Some popular international campaigns, such as “World Alzheimer’s Month”, launched in September 2012 by ADI, and “World Alzheimer’s Day”, have contributed to curbing the stigma surrounding dementia [17].

Some experimental projects targeted the spread of evidence-based messages promoting the use of physical activity and lifestyle interventions as strategies to prevent and manage AD [18]. The “Cognitive Health In Later Life” project [19] assessed the feasibility of a community-based participatory research approach to spread scientific knowledge about dementia risk factors using creative and accessible methods (e.g., street interviews, community workshops, small focus groups); this could define a working group of community members who would contribute to supporting cognitive health in their everyday lives. In another project, 100-min training program sessions were provided to family caregivers for a period of six weeks, providing education on: (i) illness issues, caring for patients living at home, and the use of health and community resources; (ii) problem-solving models; (iii) skills and strategy training [20].

(4) Mass media and creative arts could also play an important role, as the cultural representation of dementia may differ according to social context, location, and personal aging expectations.

One study [21] analyzed the communication used to talk about dementia to PLWD, by examining 350 UK national newspaper articles; it identified a “panic-blame” framework and a catastrophic lexicon juxtaposed with ambiguous lifestyle recommendations “to avoid” the disease. These findings depicted a troublesome preventable representation of dementia, shifting the responsibility for an incurable disease to a personal level.

Alzheimer’s Universe is an online educational portal related to AD management, offering evidence-based educational content [22,23].

Some experiments in creative arts have proposed “self-revelatory theater” as a resource to destigmatize dementia and to make the perception of the disease more realistic [24]. Cinema could contribute to spreading compassionate awareness, giving examples of “moral experimentation” accessible to ordinary people with a realistic portrayal of dementia. An example is the motion picture “Still Alice”, which shows how people find alternative ways to preserve their moral values, their beings, and relationships through devastating and unpredictable events, therefore becoming “researchers and experimenters of their own lives” [25].

### 3.2. Stroke

Stroke affects 15 million people worldwide each year [26]. Primary prevention is a sustainable solution for stroke [27] and requires correct and extensive public education campaigns. In 2006, the World Stroke Organization (WSO) launched the first annual “World Stroke Day”, declaring stroke a “public health emergency” and emphasizing the main stroke-related issues, such as knowledge and treatment of risk factors, stroke’s impact, and the possibility of a satisfactory life after a disabling stroke. Notably, stroke can count on famous ambassadors [28].

Surveys in high-income countries, such as Japan, revealed a lack of knowledge of the main clinical signs of stroke and emphasized the need to increase awareness [29].

The use of mass media to promote patient engagement appears to be essential to reduce post-stroke disability. One of the most important campaigns was the Act FAST Campaign [30], sponsored by the Department of Health in the UK. This acronym, FAST, identifies the main neurological deficits during a stroke (Facial weakness, Arm weakness, Speech disturbance), and emphasizes the need to promptly recognize stroke for a timely (Time) treatment. During this campaign, there was a marked increase in visualizations of stroke-related webpages, stroke-related online searches, calls to stroke association helplines, and an earlier access to emergency departments, resulting in a significant gain in quality-adjusted life years [31]. A preliminary assessment of the cost-effectiveness of the Act FAST campaign in the UK [32] concluded that every pound spent had a payback of 3.20 pounds.

Conversely, according to a subsequent analysis in the UK and Ireland, this effect was not sustained in the long term, showing a rapid decrease in the effects of this campaign [33].

Recently, the sponsored online promotion of educational material through social networks has strongly supported patients’ engagement in stroke prevention campaigns, especially among younger audiences [34]. Moreover, Twitter represents an opportunity for physicians to provide credible medical information and to engage with a greater number of patients for a more effective recognition of the early signs of stroke [35].

Stroke awareness campaigns must attract people by focusing on risk factor topics (diet, physical activities, and environmental conditions) [36]. In this regard, in 2016 the “Angels initiative” (AIN) was launched and run by the European Stroke Organization, the WSO, and Stroke Alliance for Europe [37]. It consisted of a worldwide network aimed at having 1500 “stroke-ready” hospitals in Europe by 2019, to increase the quality of stroke treatment throughout Europe. AIN not only has an educational connotation, through training courses addressed at doctors and nurses, but also is aimed at informing patients, their caregivers, and the general population about risk factors and early stroke symptoms. The success of AIN surpassed all expectations: the first three years registered 3637 hospitals with 1,350,135 treated patients. The Stroke Investigative Research and Education Network (SIREN) [38] is a further ambitious program, now active in six cities in Nigeria and Ghana, aimed at understanding genetic and environmental risk factors in Sub-Saharan Africa. SIREN seeks to characterize the genomic, sociocultural, economic, and behavioral risk factors of stroke and build effective research teams to address and reduce the burden of stroke in this area.

### 3.3. Epilepsy

Epilepsy affects around 50 million people worldwide, with nearly 80% in low- and middle-income countries [39]. In 1997, the “Global Campaign against Epilepsy: Out of the Shadows” was established to improve acceptability, treatment, services, and prevention of epilepsy worldwide. Together with the WHO, the International League Against Epilepsy, and the International Bureau for Epilepsy, it is aimed at raising “general awareness and understanding of epilepsy” and supporting Departments of Health in identifying needs and promoting education, training, treatment, services, research, and prevention in their countries. Activities include (i) educating the community to reduce the treatment gap, dispelling stigma, and the physical and social morbidity of epilepsy patients; (ii) training and educational activities for health professionals; (iii) analysis strategies for epilepsy prevention.

Since the Internet is increasingly popular with patients as a source of health information [40,41], it is not surprising that 57% of clinic-based epilepsy patients seek online information to guide their self-management [42], and that the Web is often listed among the most relevant places for information on epilepsy [43]. However, this is not enough to promote awareness, as information must be accurate and comprehensible for people with difficulty in reading and understanding written material [44]. Unfortunately, popular websites providing information on epilepsy often have a low level of readability; hence, people with low or moderate literacy skills cannot fully comprehend their content [45,46]. The free online encyclopedia Wikipedia is one of the most accessed sources on epilepsy for millions of patients worldwide [47,48], but the epilepsy-related information conveyed through Wikipedia is not always accurate. To overcome this, the International League Against Epilepsy has recently set up a joint project together with Wikipedia aimed at conveying “the most dependable and latest information on epilepsy through Wikipedia, with the widest possible penetration and range, in many languages” [48].

Analyzing online searches for epilepsy-related information can shed light on the unmet needs and the search behavior of Internet users interested in epilepsy [49,50]; it could also be useful in identifying some strategies for planning public health campaigns, to effectively promote awareness programs, increase public knowledge, and reduce stigma [47,51,52,53]. These studies consistently showed that peaks in online searches on epilepsy are often due to news of celebrities affected by seizures. Greater celebrity involvement could represent an effective strategy to increase public awareness in the public domain. The project “Stand Up For Epilepsy” by the Task Force on Sports and Epilepsy of the ILAE in collaboration with the International Bureau for Epilepsy did just this [54]. It collected photographs of famous athletes together with children or teenagers from around the world, conveying the message that “celebrities have no prejudice against the disorder and that people with epilepsy can achieve their goals, lead a full and active life and practice sports”.

### 3.4. Multiple Sclerosis

Multiple sclerosis (MS) affects approximately 2.1 million people worldwide [55].

During the “digital engagement” era, social networks (SNs) have been increasingly used [56] to exchange health information among people with MS (pwMS) [53]. Digital technology can improve knowledge about MS and facilitate clinical management for doctors and patients, although misinformation can be a serious issue and should be considered within the wider context of the so-called “post-truth era” [57,58].

However, the accuracy of online information should be carefully monitored over time and replaced by correct, evidence-based data. For instance, in 2009, chronic cerebrospinal venous insufficiency (CCSVI) was proposed as a possible cause of MS leading to a worldwide public and political controversy [59], rapidly involving mainstream media and social networks. CCSVI and its proposed treatment (i.e., widening narrowed veins in the neck and chest) swiftly gained support and triggered remarkable and relentless patient advocacy efforts. Policymakers responded differently to the public’s call for action, whereas health journalists played a key role in the media coverage of this issue. The increasing use of social media contributed to citizen engagement and advocacy. This emphasized the need to improve scientific communication, to support balanced and informed decision-making [60].

As an attempt to provide more accurate information on MS, BartsMS Blog was set up, representing a turning point in enhancing digital presence to promote quality of care, spreading valid information, and contrasting misinformation [61]. The Italian counterpart was an online community of pwMS called SMsocialnetwork.com, a Facebook-like social network, the contents of which are constantly overseen by neurologists and psychologists to avoid fake news [56].

A vast number of pwMS seek information on the Internet. However, online MS searches only partly reflect the epidemiology of the disease, and several factors can be exploited for planning health campaigns to increase awareness and fundraising. Online searches for information, however, can be driven by concurrent events that might avert public interest, independently of the true disease epidemiology [62], as recently happened in Italy, at the beginning of December 2019, when a famous rapper, Fedez, communicated to the press about his radiologically isolated syndrome, prompting a peak in searches on MS.

Another issue relating to PE in MS is the dissociation in the preferences of MS patients and current academic practice. An increasing number of pwMS read primary research publications and are involved in research and grant review activities; this should prompt a change in how studies are reported, to increase their readability to a wider audience, which now includes patients and not only researchers [63].

### 3.5. Parkinson’s Disease

Parkinson’s disease (PD) affects more than 10 million people worldwide [64].

PD study design and funding applications nowadays require the collaboration of researchers, charities, people with PD (pwPD), caregivers, and healthcare providers [65,66]. Parkinson’s UK has suggested that PE of pwPD can help researchers in: (i) identifying the most relevant gaps in the existing evidence, (ii) improving the readability of funding applications, (iii) ensuring that research is clinically meaningful, practical, and ethical for participants, (iv) supporting study conduction, and, not least, (v) divulging research results, especially for direct impact on pwPD [67,68].

Accordingly, PD celebrities can help communicate the impact of research to the public. Michael J. Fox, the iconic Hollywood actor, publicly disclosed his PD in 1998, and subsequently launched The Michael J. Fox Foundation (MJFF) for Parkinson’s research, committing himself to PD research and awareness [69]. The MJFF scientific staff contributed to the design of the Parkinson Progression Marker Initiative (PPMI), along with academic PD experts, government, and industry partners. The PPMI is a comprehensive international, multi-center study designed to identify PD progression biomarkers to improve the understanding of disease etiology, and to provide rapid public measures to enhance the development of PD-modifying therapeutic agents on their website, to accelerate research worldwide [70,71] but also for organizing local events and international publicity campaigns.

Celebrities can also increase public awareness of less common diseases. For instance, in 2014 the wife of Robin Williams disclosed that the famous actor had committed suicide after intense delusions from Lewy body disease, a type of parkinsonism [72]. This news prompted a marked increase in Google searches for information on PD and parkinsonisms [73].

Blogs can ultimately aid in integrating the needs of PD patients into clinical practice [74] and can provide evidence for regulatory and access decisions [75].

Furthermore, charities have also supported initiatives to spread research results and to showcase the complexity of PD research to the lay public, such as “Picturing Parkinson’s” and others aimed at making research accessible and comprehensible through art [76,77].

Public awareness has also benefited from World Parkinson’s Day, an opportunity for Parkinson’s communities and researchers worldwide to raise awareness, publicize efforts, and share the latest steps in research [78].

### 3.6. Migraine

Migraine affects more than a billion people all over the world, and carries huge costs for society. A survey carried out across ten EU countries (55% of the EU adult population) for The Eurolight Project found a staggering annual cost for migraine in the EU of €111 billion [79]. This PE program project provided a wealth of data on the impact of headache in Europe. The project took the form of surveys, with structured questionnaires, representing 60% of the adult EU population [80].

The recent marketing of new, efficient, and tolerable treatments [80] has increased the sources of online information on migraine available to healthcare professionals and the public, migraine patients, and their caregivers. In Healthline, [81] valuable information on migraine conditions can be found. Similarly, the “Migraine Trust” shares information on migraine treatment options, current related news, research, and practical tips for managing migraine attacks, and supports personal story-sharing [82].

Migraine apps represent an excellent tool to improve the relationship between clinician headache experts and patients and, in turn, the collection of clinical data from migraine patients. With over 2 million registered users, Migraine Buddy includes an advanced migraine diary and tracking system designed by neurologists and data scientists to help migraine patients to quickly record and identify all aspects of a migraine attack, from its triggers, symptoms, frequency, duration, pain intensity, and location, to lifestyle factors specific to each attack [83]. “Miles for Migraine” [84], founded in 2007 as a challenge for a group of researchers and clinicians dedicated to migraine and headache disorders, aimed at increasing research grants, held a first run and walk in 2008 in San Francisco’s Golden Gate Park with approximately 200 participants. Since then, various runs and walks have taken place in different US cities, such as “Migraine Education Day”, undoubtedly one of the most interesting annual PE information and experiential projects regarding migraine, for people with migraine and other headache diseases. The topics of these events cover coping strategies, advocacy, reducing stigma, research, and medical advancements.

The European Migraine and Headache Alliance [85] (EMHA) comprises over 30 patient associations for migraine and other headache diseases across Europe aiming to speak on behalf of and to advocate for the rights and needs of the 138 million people in Europe living with headache. Through EMHA, a project with the evocative name of “My Migraine Voice” has been conducted with 11,266 participants from 31 countries, constituting to date the largest survey conducted in migraine patients [86].

### 3.7. Neurogenetics

Neurogenetic disorders are a wide and heterogeneous group of conditions caused by changes in one or more genes or chromosomes. Rare diseases are a global public health concern, with an estimated prevalence worldwide of 350 million individuals. Although the number of patients with each specific rare disease is limited, collectively, rare diseases affect about 30 million people in the United States and 30 million in Europe [87]. Their management is a scientific and medical challenge because of limited research funding compared with more common diseases, and difficulty in tracing and enrolling patients in clinical studies [88].

There exists the need to spread awareness of rare diseases, at a scientific level for healthcare professionals, and at a public health level for patients and their caregivers, but also for the general population. In this field, PE can promote drug development, disease management, and fundraising for research and support.

Patient organizations (POs) play an important role in addressing and overcoming the challenges of patient management and drug development [88]. Consequently, many patient advocacy organizations today are led by professionals and thus have a large impact on research [89,90].

POs mainly concentrate on educating patients, physicians, and the community about disease and management and treatment innovation; championing and directly funding efforts to increase disease comprehension and develop new therapies; forming connections between disease key opinion leaders and drug companies; and providing drug developers with relevant insights into the patient community to facilitate the development of therapies that best meet the community’s needs [91,92].

This involvement, also known as engagement in research, occurs when patients and caregivers collaborate in prioritizing, creating, conducting, and spreading research [93]. Over the last decade, policies and initiatives encouraging patient engagement in healthcare and research have become a strategy to make the healthcare system more centered on patients and thus more resourceful with better health outcomes [94]. A clear example is Parent Project Muscular Dystrophy, which launched a new Duchenne registry app, aimed at enhancing patients’ engagement and resources. Data collected with this app will then be stored in a data bank to combine data from The Duchenne Registry with data from many hospitals in the United States, post-marketing surveillance data provided by industry partners with approved therapies, and datasets from academic and other advocacy partners [95].

Public exhibitions are a further way of providing information to the community about a rare neurogenetic disease. In Italy, a good example is the exhibition entitled “secondo nome Huntington”, promoted by the Huntington ONLUS, an association whose mission is to inform and make the community aware of delicate aspects of this complex disease [96].

The Queen Square Centre for Neuromuscular Diseases has organized initiatives such as the Mitochondrial Disease Patient Engagement Day 2018 or the Charcot Marie Tooth Patient Day 2017 [97].

Neurofibromatosis (NF) is one of the world’s most common genetic conditions, and the campaign “Neurofibromatosis Awareness Day” can spread information about the disease to attain greater acceptance.

The Children’s Tumor Foundation [98] provides patient and family support thanks to its information resources, youth programs, and local chapter activities; it can help in finding effective treatments and access to quality patient healthcare through its national NF Clinic Network.

The National Institute of Neurological Disorders and Stroke of the National Institutes of Health through the Neurofibromatosis Fact Sheet [99] have also provided information for patients about the different features of the different types of NF, new treatments, prognosis, prevention through prenatal diagnosis, and how to get involved in research programs.

For patients affected by tuberous sclerosis (TSC), The Tuberous Sclerosis Association (TSA) [100] was launched in 1977 and has become a professional organization providing support to families affected by TSC across the UK.

TSA is a registered charity with three charitable objectives: (i) support TSC individuals, together with their families or carers; (ii) encourage and support research into the causes and management of TSC; (iii) provide knowledge and information.

To improve research in this field, the TS Alliance Unveils New Research Business Plan [101] has launched a new, comprehensive five-year research business plan. Building on the deep relationships with both the TSC patients and research communities, its aim is to create resources to help scientists to find a cure and improve the course of this disease.

### 3.8. Amyotrophic Lateral Sclerosis

Amyotrophic lateral sclerosis (ALS) is a rare neurodegenerative disease [102] affecting about 420,000 of the world’s population.

Over the last few decades, sharing and spreading knowledge of ALS through the disease experience of celebrities affected by ALS has increased public awareness. Notably, ALS is also known as “Lou Gehrig’s disease” after the famous baseball player Lou Gehrig, who delivered his farewell speech in 1939, just 2 months after learning he had ALS [103]. His speech was impressive, and Gehrig saw himself “not as a mere victim of a form of paralysis but a symbol of hope for thousands of sufferers of the same disorder” [104], making a profound impact on public ALS awareness in the USA and beyond, prompting clinicians and researchers to fight against ALS worldwide.

Many other famous people with ALS have engaged the “non-scientific public” community with the everyday problems caused by the disease and have collaborated with researchers in supporting public initiatives aimed at funding research projects and spreading updated information on ALS pathogenesis and management. In this regard, professional Italian soccer players who have suffered from ALS, such as Stefano Borgonovo, founder of the “Borgonovo Foundation”, collaborate with researchers to fund research and corroborate ALS public awareness. Interestingly, several studies have investigated the association between soccer and ALS: Chiò et al. [105] performed the first retrospective study in a large cohort of male professional soccer players in the period 1970–2001, concluding that soccer increased the risk for ALS. These findings had a wide impact within the Italian sports and scientific community and, subsequently, sports newspapers and news agencies released the news of 51 male soccer players reportedly with ALS [106]. More recently, a survey analysis on male Italian soccer players affected by ALS, and cited either on the Internet or in journalists’ books [107], confirmed the existence of an ALS cluster among Italian professional soccer players, requiring further research to explain the reasons for this unexpected and still unintelligible association.

Public interest in ALS has grown tremendously [108], especially after the “Ice Bucket Challenge” (IBC), a Web event that took place in August 2014: after jumping into or getting doused with cold water on videos that were posted to social media, people would challenge others to do the same and make a donation to charity. Pete Frates, an ex-baseball player affected by ALS, launched this initiative, which went viral, especially through social networks such as Twitter and Facebook. The IBC raised $220 million involving more than 17 million people worldwide, from high profile celebrities, politicians, and researchers to people from the poorest regions of the globe. Unfortunately, by September 2014 donations had dropped to near pre-challenge levels [109], although ALS public awareness created by the IBC lasted in the long term, with an increase in ALS-related literature [110]. The IBC phenomenon taught us that the appropriate use of social media might represent a key step in engaging public awareness and prompting research into rare diseases [111]. With regard to PE in genetic research on ALS, the pressing need to study rare variation across the full length of the genome in cases with and without family history of ALS prompted the creation of Project MinE (meaning the “mine” of genetic data), a large-scale whole-genome sequencing study on ALS, based on a Consortium design which included research groups and funding agencies [112]. Project MinE aims to obtain sequencing data on 15,000 ALS patients and 7500 matched controls associated with ALS risk, fine-map known and novel loci, and provide a publicly available summary-level dataset that could enable further genetic research on this and other rare diseases. It represents a robust example of data sharing and transparency in scientific research, useful for: analyzing large sets of samples, ensuring rigorous experiments that can be reproduced by external groups, and allowing publicly funded research to be made available to the public itself. In particular, genotype frequency information and genic burden testing results are publicly available at the project’s online browser (http://databrowser.projectmine.com/; accessed on 26 March 2021). This browser has been continuously updated as the project expanded, and has allowed the sharing of data and results with the scientific and healthcare communities, as well as the public more broadly. Currently, the Project MinE initiative has sequenced 4366 ALS patients and 1832 age- and sex-matched controls [113].

Interaction between researchers and ALS patients has also helped make patients aware of ongoing clinical trials and has facilitated their enrolment. Stronger partnerships between patients, clinicians, government, non-profit organizations, and regulatory agencies will significantly affect treatment development [114]. Remarkably, crucial items for patients with ALS are also the choice of palliative care and the management of the end of life. Challenging debates with ALS patients include the use of percutaneous endoscopic gastrostomy and/or assisted ventilation, end-of-life planning, and requests for assisted dying [115]. Physicians’ attitudes toward end-of-life decisions in ALS vary greatly according to patient-related features (i.e., prognosis, phenotype), personal characteristics (i.e., creed, palliative care training), and physicians’ role in the decision-making process (i.e., truth-telling, informed consent) [116]. Considering that clinical recommendations and healthcare law regarding end-of-life decision planning in ALS are still a major subject of study and debate, clinical experts have been more frequently engaged in consensus conferences on this issue to guarantee effective communication and implementation of advanced directives, recognized as paramount to preserving patient autonomy and dignity [117]. Psychological factors, education level, and cognitive status (especially the occurrence of executive dysfunction) have a key role in end-of-life decisions of patients with ALS or other neurological diseases. Moreover, advanced care planning should be recommended in anticipation of emergency interventions. These decisions should be dis-cussed by healthcare professionals and patients, based on the wishes of the patients and caregivers, and communicated to all healthcare professionals who are involved in the management of ALS patients, also including the medical and social care networks employed in disability support and home care. Therefore, the social and familial engagement in palliative care should work in partnership with ALS centers through information-sharing and collaborative discussions [118]. 

## 4. Discussion and Conclusions

PE in neurological disorders is rapidly improving and can be a valid tool with numerous applications, from patients’ engagement in research policies to dedicated social network development for communication between scientists and patients, and from infodemiology analyses to decisions regarding disease management (Table 2).

To date, different approaches have been used to encourage the dissemination of scientifically correct information about different neurological diseases among laypeople. Over the last few decades, celebrity endorsement has been a very powerful tool in health campaigns for engaging public awareness and funding research on neurological diseases, especially through powerful social networks publicizing science. The growing interaction among clinicians, researchers, government, and non-profit organizations should bear the responsibility for taking on the substantial task of promoting the development of a system for medical care and research on neurological diseases. However, further efforts are still needed to communicate crucial items to the public, especially regarding the clinical trial status and end-of-life care, and to promote the use of social media for further PE scientific activities. Moreover, clinicians are often under pressure and bear a high responsibility to face the understandable search for “miraculous” therapies, which are sometimes required by patients and caregivers. As such, patients try to obtain prescription of non-approved, non-effective, and sometimes harmful medications and procedures, as it was in the case of chronic cerebrospinal venous insufficiency (CCSVI) in MS [119]. We should then engage with the public to ensure that the uptake and spread of scientific discoveries via social media are viewed and interpreted in an appropriate context.

Another significant PE contribution based on the wide accessibility to the Internet is the change in recruitment for clinical trials and observational studies, enabling people with neurological diseases living in rural and remote communities to be included as well [120]. Furthermore, as shown by the cost-effectiveness assessment of the Act FAST campaign in the UK, PE, when successful, could guarantee a positive payback.

Further improvements in PE should include not only a careful commitment to finding attractive topics to divulge but also the provision of accurate and easy to understand information for people with difficulty in reading and understanding written material; in this regard, videos could be a simple way to convey information.

As the dissemination of news on the Web has a transitory effect, it would be useful to provide a periodic recall of public attention. An increasing number of people seek information on their neurological disease on the Internet, thus generating a data flow that can be analyzed for research and used for health policy purposes, but also to understand the unmet needs of patients better and to thus improve their quality of life.

Lastly, we should become aware that promoting well-being and optimal mental health is just as important as promoting knowledge about mental illnesses. For instance, complex but also enjoyable PE activities that can be structured over time (e.g., interdisciplinary Open University courses teaching participants how to develop positive relationships and engage in interesting activities) could be proposed as a tool for improving well-being, thus contributing to preventing mental illness [120].

## Figures and Tables

**Figure 1 brainsci-11-00429-f001:**
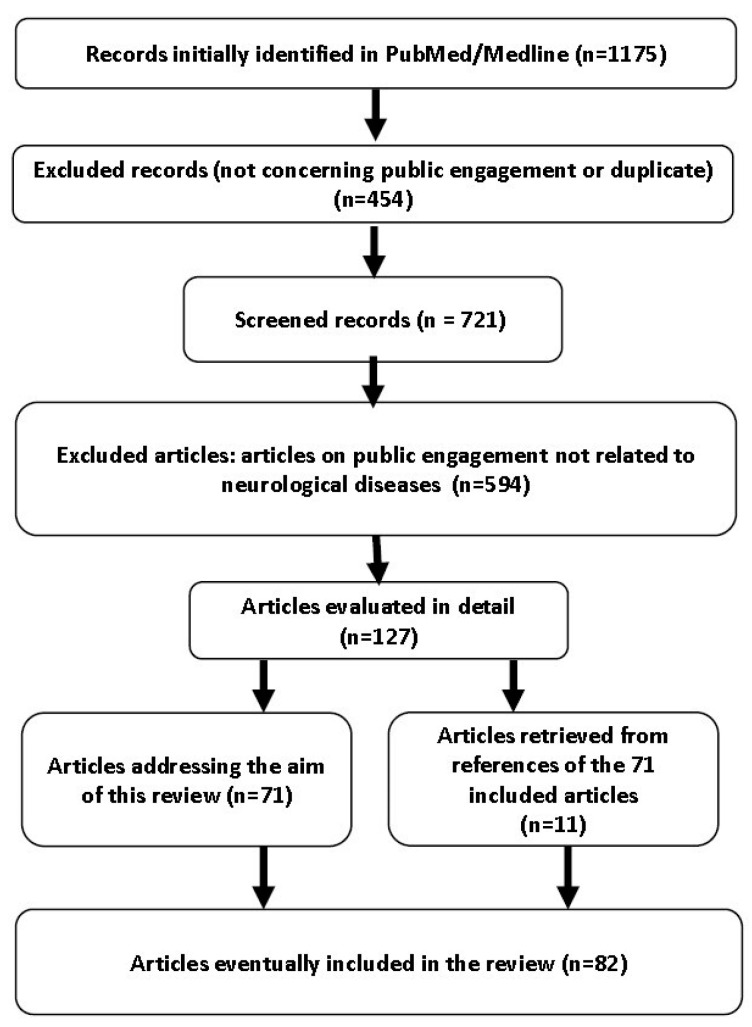
Flow chart of article selection.

**Table 1 brainsci-11-00429-t001:** Ways of providing information on processes and results of research [1].

Informative research events open to the public (e.g., Researchers’ Night, Awareness Day)
Project communications
School courses, orientation and interaction initiatives with high schools
Popular publications in national or international magazines written by researchers
Participation of researchers in national and international radio and television broadcasts
Active researcher participation at public meetings (e.g., scientific cafés, scientific festivals, etc.)
Interactive and/or informative websites, blogs, forums, social networks, Web communities, etc.
Open days at museums, libraries, theaters, university buildings
Organization of concerts and exhibitions
Health protection initiatives (e.g., information and prevention days)

**Table 2 brainsci-11-00429-t002:** Identified aims of public engagement for neurological diseases.

Provide good-quality, timely, and easily accessible health-related information
Provide opportunities for patients and their families to connect, support, and learn together
Increase the knowledge and expertise of professionals working with patients
Support the research agenda to improve outcomes and quality of life
Expand the understanding of neurological disorders to improve diagnosis and development of new treatments
Disseminate evidence to positively influence National Health Service (NHS) policies and practice
Spread informative campaigns for better access to NHS treatments and care
Promote the development of centers of excellence to support patients and their families
Pilot and bring forward innovative approaches to person, family, and carer support
Emphasize the importance of research for obtaining advancements in the development of new therapeutic approaches
Emphasize the importance of prevention through appropriate lifestyles

## Data Availability

Not applicable.

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
