# Peer review of "Public Engagement and Neurology: An Update"

_brainsci, 2021, doi:10.3390/brainsci11040429_

Round 1
Reviewer 1 Report
Thank you for an interesting manuscript.
The text include numerous web links. These need to be cited appropriately.
Please add relevant details to the methods to enable them to be repeatable e.g. please add the inclusion and exclusion criteria. Suggest to make a flow chart that shows how many papers where initially identified and how many papers were rejected and how many were included, etc. Please present the search terms using the Boolean system.
You haven’t discussed the results. Suggest to expand your conclusions section, name it as Discussion and conclusions.
Author Response
Thank you for an interesting manuscript.
- The text include numerous web links. These need to be cited appropriately.
We thank the reviewer for this comment and suggestion. In the revised manuscript we have provided the web links in the references.
2) Please add relevant details to the methods to enable them to be repeatable e.g. please add the inclusion and exclusion criteria.
We thank the reviewer for this comment. In the revised version we have provided relevant details to ensure repeatability. The text was modified as follows (page 3):
In this review we included articles specifically addressing the role of PE in the following neurological disorders or conditions: Dementia, Stroke, Epilepsy, Multiple Sclerosis, Parkinson’s disease, Migraine, Neurogenetics, and Amyotrophic Lateral Sclerosis. We excluded articles on PE not related to the aforementioned neurological diseases. For the assessment of articles for possible inclusion we adopted the definition of PE provided by the England’s University funding agency as reported above [1]. To assist reviewers in the article selection, we provided some examples of PE (Box 1).
An advanced literature search in MEDLINE (accessed through PubMed) was conducted at the beginning of 2020 by two independent Authors (GA, SE). The search terms were “(public engagement OR third mission) AND (motoneuron disease OR amyotrophic lateral sclerosis OR alzheimer OR dementia OR multiple sclerosis OR epilepsy OR stroke OR Parkinson disease OR headache OR migraine OR neurology and genetic)”. We ordered the results according to the “best match” filter. The search was restricted to articles from peer-reviewed journals published in English. No date restrictions were used. The repeatability of the data selection method between the 2 reviewers was tested using a stepwise searching approach. Starting from the research with the main number of results ("(Dementia OR Alzheimer) AND (public engagement OR third mission)"), the first 100 results of 540 were independently evaluated by the two reviewers. Therefore, any disagreement was discussed to clarify selection criteria. Agreement on screening decision was > 95%, so the further selections regarding other combinations were performed by one reviewer only (GA). The reference lists of the included studies were manually searched to identify further relevant studies.. A detailed flow-chart of article selection is presented in Figure 1.
Moreover, based on the PE activities formally identified for qualitative analyses by the Department of Advanced Medical and Surgical Sciences of the University of Campania “Luigi Vanvitelli”, we also carried out a search on Google for web pages, online communities, forums, sites and Institutional websites, using the following terms (even in combination): “Alzheimer disease”, “dementia”, “stroke”, “epilepsy”, “multiple sclerosis”, “Parkinson disease”, “migraine”, “headache”, “Huntington disease” “neurofibromatosis”, “muscular dystrophy”, “tuberous sclerosis”, “rare neurological disease”, “amyotrophic lateral sclerosis”, “public engagement” “awareness day”, “day”, “open day”, “fundraising”, “patients’ associations”, “research engagement”, “parent project”, "exhibition", “onlus”, “engagement day”, “campaign”, “foundation”, “alliance”, “ celebrities”, “viral event”, “non-profit organization”, “prevention”, “museum”, “sport”. Relevant results from Google search were reported.
3) Suggest to make a flow chart that shows how many papers where initially identified and how many papers were rejected and how many were included, etc. Please present the search terms using the Boolean system.
We thank the reviewer for this suggestion. We have now included a flow chart (Figure 1) showing how many papers where initially identified and how many papers were rejected and how many were included. In the added details requested in the previous comment, we have provided the search terms using the Boolean system (page 3).
4) You haven’t discussed the results. Suggest to expand your conclusions section, name it as Discussion and conclusions.
We have changed the name of this section, and expanded the discussion by adding a paragraph on mental illnesses, and referring to the responsibilities of physicians towards prescription of “miraculous therapies” with little or no evidence. The main aims of PE, as identified from the selected and included articles, was listed in Box 2 (page 13-18).

Reviewer 2 Report
Very interesting and relevant work addressing the public engagement in different neurological fields.
Nevertheless, it would be important to clarify the following:
- Being a "systematic review", as mentioned by the authors, which inclusion/exclusion criteria were used for selecting the articles and how was the analyses of the selected articles carried out? In this line, could important public engagement have been missed (for example, why was the project MinE not considered, regarding ALS or neurogenetics)?
In addition
- How do the authors see the pressure of the patients and patient associations into the prescription of non-approved, non-effective medications?
- How do the authors see the social and familiar engagement when addressing the complex questions that were mentioned, as end-of life decisions?
Author Response
Very interesting and relevant work addressing the public engagement in different neurological fields.
Nevertheless, it would be important to clarify the following:
5) Being a "systematic review", as mentioned by the authors, which inclusion/exclusion criteria were used for selecting the articles, and how was the analyses of the selected articles carried out?
As requested also by reviewer 1, in the revised manuscript we have described in the methods section the procedure of articles selection and the inclusion/exclusion criteria.
6) In this line, could important public engagement have been missed (for example, why was the project MinE not considered, regarding ALS or neurogenetics)?
We thank the reviewer for this valuable comment. In the revised version we have mentioned the characteristics of the project MinE’s and its aims, as a pertinent example of a research project engaging research groups, funding agencies and the public.
The text was modified as follows (page 11, lines 474-491):
With regard to the PE in genetic research on ALS, the pressing need to study rare variation across the full length of the genome in cases with and without family history of ALS prompted to begin the Project MinE (meaning the “mine” of genetic data), a large-scale whole-genome sequencing study in ALS, based on a Consortium design which included research groups and funding agencies [112] Project MinE aims to obtain sequencing data in 15,000 ALS patients and 7,500 matched controls associated to ALS risk, fine-map known and novel loci, and to provide a publicly-available summary-level dataset that could enable further genetic research of this and other rare diseases. It represents a robust example of data sharing and transparency in scientific research, useful for: analyzing large sets of samples; ensuring rigorous experiments that can be reproduced by external groups; and allowing for publicly-funded research to be made available to the public itself. In particular, genotype frequency information and genic burden testing results are publicly available at the project’s online browser (http://databrowser.projectmine.com/). This browser has been continuously updated as the project expanded, and has allowed sharing data and results with the scientific and healthcare communities, as well as the public more broadly. Currently, the Project MinE initiative has sequenced 4,366 ALS patients and 1,832 age- and sex-matched controls [113].”
In addition
7) How do the authors see the pressure of the patients and patient associations into the prescription of non-approved, non-effective medications?
We thank the reviewer for giving us the opportunity to address this issue. In the conclusions we have added a paragraph related to this issue.
The text was modified as follows (page 14, lines 543-549):
Moreover, sometimes clinicians are often under pressure and bear a high responsibility to face the understandable search for “miraculous” therapies, that are sometimes required by patients and caregivers. As such, patients try to obtain prescription of non-approved, non-effective and sometimes harmful medications and procedures, as it was in the case of Chronic Cerebrospinal Venous Insufficiency (CCSVI) in MS [119]. We should then engage with the public to ensure that uptake and spread of scientific discoveries via social media are viewed and interpreted in an appropriate context.
8) How do the authors see the social and familiar engagement when addressing the complex questions that were mentioned, as end-of life decisions?
We thank the Reviewer for this suggestion. We have discussed this relevant issue in the revised version of the text (page 12, lines 507-517):
Psychological factors, education level and cognitive status (especially the occurrence of executive dysfunction) have a key role in end-of-life decisions of patients with ALS or other neurological diseases. Moreover, advanced care planning should be recommended in anticipation of emergency interventions. These decisions should be discussed by healthcare professionals and the patients, and based on the wishes of the patients and caregivers, and communicated to all healthcare professionals who are involved in the management of ALS patients, also including the medical and social care networks employed in disability support and home care. Therefore, the social and familial engagement in palliative care should work in partnership with ALS centers through information-sharing and collaborative discussions [118]. We integrated this comment in the revised text.

Reviewer 3 Report
The authors conducted a review of the literature attempting to evaluate the role of Public Engagement (PE) in different neurological conditions.
The review is very rich and full of interesting details and giving an account of the current landscape, and its evolution, in particular thanks to social networks and the Internet.
The paper is generally well written and structured.
However, I have several remarks on the text, as well as additional suggestions.
Given these items the manuscript requires minor revisions.
- The authors introduced the concept of Public Engagement (PE) in the very first lines of the introduction. They indicated that this concept dated back to 2006 but did not directly specify the source there in the paper. It would be necessary to specify: (as defined by England’s university funding agency, HEFCE, in 2006).
- The authors list the keywords used in their database queries. Inserting a table containing the words would clarify the text.
- Box2 is not mentioned in the text
- The last lines of the conclusion are unclear and especially the link with the concept “Flourishing” is not obvious. “Lastly, complex but also enjoyable PE activities that can be structured over time (e.g., interdisciplinary Open University courses) could be proposed as a tool for “training” the brain while apprehending new information, thus becoming in themselves a preventive approach for many healthy people [82]”. Flourishing: An Evolutionary Concept Analysis. “Flourishing refers to the experience of life going well. It is a combination of feeling good and functioning effectively. Flourishing is synonymous with a high level of mental well-being, and it epitomizes [sic]mental health.”
- Numerous web links do not target the expected page
L43 : (https://ec.europa.eu/programmes/hori-43 zon2020/en/h2020-section/public-engagement-responsible-research-and-innovation)
L46: (https://www.euro.who.int/en/data-and-evi-46 dence/news/news/2016/09/what-is-the-burden-of-disease-in-the-region).
L63: (https://www.damss.unicampania.it/terza-63 missione),
L139: (www.strokecenter.org/pa-139 tients/about-stroke/stroke-statistics)
L170: (www.angel-initiative.com)
L208: (www.ilae.org/jour-208 nals/ilae-wikipedia).
L218: (https://www.ilae.org/about-218 ilae/public-policy-and-advocacy/epilepsy-and-sport-project-stand-up-for-epilepsy)
L274: (https://www.michaeljfox.org/michaels-274 story).
L295: (https://www.parkinsonsmove-295 ment.com/world-parkinsons-day-2019/).
L306: (https://www.health-306 line.com)
L366: (https://www.prnewswire.com).
L374: (https://www.ucl.ac.uk/centre-for- neuromuscular-diseases/patient-and-374 public-engagement).
Author Response
The authors conducted a review of the literature attempting to evaluate the role of Public Engagement (PE) in different neurological conditions.
The review is very rich and full of interesting details and giving an account of the current landscape, and its evolution, in particular thanks to social networks and the Internet.
The paper is generally well written and structured.
However, I have several remarks on the text, as well as additional suggestions.
Given these items the manuscript requires minor revisions.
9) The authors introduced the concept of Public Engagement (PE) in the very first lines of the introduction. They indicated that this concept dated back to 2006 but did not directly specify the source there in the paper. It would be necessary to specify: (as defined by England’s university funding agency, HEFCE, in 2006).
As asked by the reviewer, we have modified the first paragraph of the Introduction as follows (page 2, lines 34-37):
As defined by the England’s University funding agency, HEFCE, in 2006, Public Engagement (PE) is the involvement of “specialists who listen, develop their understanding and interact with non-specialists" in non-profit activities of educational, cultural and social nature to engage the public in science-related matters”.
10) The authors list the keywords used in their database queries. Inserting a table containing the words would clarify the text.
Thanks for pointing this out. In the revised manuscript, as also asked by reviewer 1, we added in the methods section the search terms used for Pubmed and Google search (See page 3), therefore we did not insert a table.
An advanced literature search in MEDLINE (accessed through PubMed) was conducted at the beginning of 2020 by two independent Authors (GA, SE). The search terms were “(public engagement OR third mission) AND (motoneuron disease OR amyotrophic lateral sclerosis OR alzheimer OR dementia OR multiple sclerosis OR epilepsy OR stroke OR Parkinson disease OR headache OR migraine OR neurology and genetic)”. We ordered the results according to the “best match” filter. The search was restricted to articles from peer-reviewed journals published in English. No date restrictions were used. The repeatability of the data selection method between the 2 reviewers was tested using a stepwise searching approach. Starting from the research with the main number of results ("(Dementia OR Alzheimer) AND (public engagement OR third mission)"), the first 100 results of 540 were independently evaluated by the two reviewers. Therefore, any disagreement was discussed to clarify selection criteria. Agreement on screening decision was > 95%, so the further selections regarding other combinations were performed by one reviewer only (GA). The reference lists of the included studies were manually searched to identify further relevant studies.. A detailed flow-chart of article selection is presented in Figure 1.
Moreover, based on the PE activities formally identified for qualitative analyses by the Department of Advanced Medical and Surgical Sciences of the University of Campania “Luigi Vanvitelli”, we also carried out a search on Google for web pages, online communities, forums, sites and Institutional websites, using the following terms (even in combination): “Alzheimer disease”, “dementia”, “stroke”, “epilepsy”, “multiple sclerosis”, “Parkinson disease”, “migraine”, “headache”, “Huntington disease” “neurofibromatosis”, “muscular dystrophy”, “tuberous sclerosis”, “rare neurological disease”, “amyotrophic lateral sclerosis”, “public engagement” “awareness day”, “day”, “open day”, “fundraising”, “patients’ associations”, “research engagement”, “parent project”, "exhibition", “onlus”, “engagement day”, “campaign”, “foundation”, “alliance”, “ celebrities”, “viral event”, “non-profit organization”, “prevention”, “museum”, “sport”. Relevant results from Google search were reported.
11) Box2 is not mentioned in the text
Thanks. We have mentioned Box 2 in the main text (page 13, line 532).
12) The last lines of the conclusion are unclear and especially the link with the concept “Flourishing” is not obvious. “Lastly, complex but also enjoyable PE activities that can be structured over time (e.g., interdisciplinary Open University courses) could be proposed as a tool for “training” the brain while apprehending new information, thus becoming in themselves a preventive approach for many healthy people [82]”. Flourishing: An Evolutionary Concept Analysis. “Flourishing refers to the experience of life going well. It is a combination of feeling good and functioning effectively. Flourishing is synonymous with a high level of mental well-being, and it epitomizes [sic]mental health.”
We thank the reviewer for these comments, that we integrated in the revised version of the manuscript. The text was modified as follows (page 14, lines 565-570):
Lastly, we should become aware that promoting well-being and optimal mental health is as much important as promoting knowledge about mental illnesses. For instance, complex but also enjoyable PE activities that can be structured over time (e.g., interdisciplinary Open University courses teaching how to develop positive relationships and engage in interesting activities, could be proposed as a tool for improving well-being, thus contributing to prevent mental illness [120].
14) Numerous web links do not target the expected page
Thanks for pointing this out. We have verified that each web link targets the expected page. As required by reviewer 1 we have listed them in the reference list.
L43: (https://ec.europa.eu/programmes/hori-43 zon2020/en/h2020-section/public-engagement-responsible-research-and-innovation)
L46: (https://www.euro.who.int/en/data-and-evi-46 dence/news/news/2016/09/what-is-the-burden-of-disease-in-the-region).
L63: (https://www.damss.unicampania.it/terza-63 missione),
L139: (www.strokecenter.org/pa-139 tients/about-stroke/stroke-statistics)
L170: (www.angel-initiative.com)
L208: (www.ilae.org/jour-208 nals/ilae-wikipedia).
L218: (https://www.ilae.org/about-218 ilae/public-policy-and-advocacy/epilepsy-and-sport-project-stand-up-for-epilepsy)
L274: (https://www.michaeljfox.org/michaels-274 story).
L295: (https://www.parkinsonsmove-295 ment.com/world-parkinsons-day-2019/).
L306: (https://www.health-306 line.com)
L366: (https://www.prnewswire.com).
L374: (https://www.ucl.ac.uk/centre-for- neuromuscular-diseases/patient-and-374 public-engagement).

Round 2
Reviewer 2 Report
The authors have answered all my concerns adequately. I have no further comments.